# Examining the attitudes of sexually abused and non-abused individuals towards marriage in terms of ambivalent sexism

**Eyüp Çelik**[1]*, **Kübra Dombak**[2], **Mehmet Kaya**[1], **Ümit Sahranç**[1], **Samet Makas**[1], **Lokman Koçak**[3], **Mithat Takunyacı**[1], **Seyhan Bekir**[4]

**1** Faculty of Education, Sakarya University, Sakarya, Turkey, **2** Institute of Education Sciences, Sakarya University, Sakarya, Turkey, **3** Faculty of Education, Bayburt University, Bayburt, Turkey, **4** Institute of Education Sciences, Uludag University, Bursa, Turkey

☉ These authors contributed equally to this work.
* eyupcelik@sakarya.edu.tr

## Abstract

The research aims to examine the attitudes of individuals who are victims of abuse and those who are not towards marriage in terms of ambivalent sexism. The research study group consists of 718 individuals between the ages of 18–48. Research data were collected with the Inonu Marriage Attitude Scale and Ambivalent Sexism Inventory. As a result of the correlation analysis, it was concluded that the marriage attitude was positively and significantly correlated with hostile and protective sexism. However, since the relationship between hostile sexism and attitudes towards marriage is lower than that of protective sexism, hostile sexism was not included in the model as a control variable. In the covariance analysis, it is seen that protective sexism and sexual abuse predict the attitude towards marriage at a statistically significant level. In addition, when the effect of sexual abuse on the attitude towards marriage was examined by controlling the protective sexism variable, it was found that it was statistically significant without the effect of sexism. According to the findings, it was determined that individuals who were not victims of sexual abuse had higher attitudes towards marriage than those who were victims.

## Introduction

All kinds of sexual crimes, from verbal abuse to rape, are as old as human history. Sexual abuse is unwanted sexual acts that occur when the perpetrators use force, threaten or take advantage of individuals who do not consent, under the influence of inappropriate lustful desires [1]. These actions include non-contact actions such as exhibitionism, voyeurism, obscene speeches, or watching pornographic films, as well as actions that can occur with contact in the form of sexual touch, genital contact, or penetration [2]. When the literature is examined, there are many studies on sexual abuse [3–9]. According to the results of the studies, sexual abuse is experienced by individuals of all ages, and it is more common, especially in children and adolescent girls [10]. According to the statistical data obtained from social studies in the

**Funding:** The authors received no specific funding for this work.

**Competing interests:** The authors have declared that no competing interests exist.

USA, 81% of women and 43% of men have been exposed to sexual harassment and/or sexual assault in their lifetime, and a quarter of men and one-third of women have a history of first sexual abuse between the ages of 11 and 17 [11]. The vast majority of cases occur within the family or by people close to the family [12]. Studies reveal that sexual abuse effects people in term of various way, such as well-being and psychosocial functioning including negative impacts on sleep [13], dissociation and suicide attempts [14], neuropsychological deficits, specifically executive function and IQ deficits [15], stress on depression symptoms [16], and attention deficit hyperactivity disorder [17].

After sexual abuse, the victim may experience some cognitive, emotional, and behavioral symptoms, as well as various relationship problems, brought about by feelings such as insecurity and disappointment [18,19]. According to a study, it is stated that after abuse, victims perceive their partners more negatively, experience problems such as distrust towards their partners and not enjoying sexual intercourse, and especially victims of rape are dissatisfied and unhappy in their relationships [20,21]. Furthermore, Wang et al. found that there is the relationship between child sexual abuse and sexual dysfunction in adults [22]. When all symptoms that may occur after abuse are evaluated, the victim's perception of people and the environment may change with the effect of the trauma he/she has experienced. As a result of this situation, it can be said that the individual may evaluate the romantic relationship and the institution of marriage negatively, and therefore her attitude towards marriage may be affected.

The belief and attitude that an individual develops towards marriage include the knowledge and experiences he/she has acquired throughout his/her life, regardless of his/her past experiences and traumas [23]. In this context, marital attitude includes the individual's feelings, thoughts, and behaviors towards the marital relationship or the institution of marriage. A person's positive or negative attitude towards marriage is shaped by a number of factors ranging from the characteristics of the society and common attitudes towards marriage to the experience in one's own family [24]. In addition to these, age [25], gender [26], religious belief [27], economic status [28], parental union [29], being in a romantic relationship [30], social media, and role models such as married relatives and friends are among the factors that will affect people's attitudes towards marriage.

Due to changes in marital relationships in recent years, research focusing on marriage attitudes and marriage plans of young adults and adolescents has become more common [31–36]. When the literature is examined, some studies examining the relationship between marriage attitude and gender [37,38] reveal that women develop more positive attitudes than men, while some studies [26] showed that men have more positive attitudes towards marriage. When the relationship between marriage attitude and family structure [39] was examined, it was found that the attitudes of people who grew up in a conflicted family environment towards marriage were also negative. Similarly, it has been determined that individuals whose parents are divorced exhibit more negative attitudes than those whose parents are not divorced [29]. In this context, it is seen that gender is an important factor on marriage attitude, like intrafamilial conflicts and parental union, but there is no definite difference between genders. It can be said that this variability between men and women is related to the roles that society assigns to women and men in the institution of marriage and to gender stereotypes. Different role expectations, stereotypes, and gender perceptions between the sexes can be thought of as related to the concept of sexism.

Sexism is defined as making one sex more important and superior than the other, discriminating between the sexes [40]. However, sexism does not always consist of negative judgments. Studies show that not only negative behaviors towards women but also positive attitudes and behaviors that keep women in a superior position are sexism [41]. This situation related to the

ambivalent sexism which is the coexistence of men's structural power (control over political, economic, legal, and religious concepts) and women's dyadic power (which makes men dependent on women romantically and sexually in interpersonal relations and empowers women against men) [42]. According to the theory, ambivalent sexism consists of two main dimensions: hostile sexism and protective sexism [43]. Hostile sexism includes attitudes that emphasize that women are dependent, weak, and in the background and that negative attitudes towards women manifest themselves clearly. Protective sexism, on the other hand, is the legitimization of male dominance and the similar function with hostile sexism by positioning women at a lower level than men, although it refers to positive attitudes and behaviors such as protecting, caring for, loving, and glorifying women [44].

As a result, marriage has an important position in the formation of the family, which is one of the cornerstones of society. In order for a healthy marriage relationship to be established, individuals must first have positive attitudes towards marriage. However, the past life and traumas of the person together with many variables can affect the marriage attitude. Sexual abuse, which is a type of trauma, can also change the individual's perspective towards marriage. Healthy individuals make up healthy families, and healthy families make up healthy individuals. From this point of view, it is thought that traumatic experiences such as sexual abuse may have effects on the individual's attitudes towards marriage and the family institution he will establish in the future. As can be understood from related studies that sexual abuse is related to marriage attitudes [24,45]. However, there may be other variables that will affect the relationship between sexual abuse and marital attitude. When the studies on abuse and marriage were examined, it was seen that other variables that could affect the relationships between these two variables were not studied. One of these variables may be sexism in individuals with a history of sexual abuse, due to the perception that women should be protected, or because of the feelings of worthlessness after sexual abuse, where women may be secondary to men at home and in working life. Therefore, in this study, it is aimed to examine the attitudes of individuals who are victims of abuse and those who are not, towards marriage in terms of ambivalent sexism.

## Materials and methods

This research was conducted based on the correlational research model, one of the quantitative research methods. Ethics approval was received from the Ethics Committee of Sakarya University (Ethical Application Ref: E-61923333-050.99–33880). Informed written consent was obtained from all individual participants included in the study.

### Study group

Research data were collected from a total of 718 people, 471 (65.6%) women, and 247 (34.4%) men, with an average age of 22.5 years. Of the participants who voluntarily participated in the study, 257 (35.8%) were victims of sexual abuse, and 461 (64.2%) were individuals who were not victims of sexual abuse.

### Data collection tools

**Inonu Marriage Attitude Scale (IMAS).**    The scale, developed by Bayoğlu and Atli, is a one-dimensional, 21-item, 5-point Likert-type (1 = strongly disagree, 5 = strongly agree) measurement tool. The marriage attitude score is obtained by summing the scores obtained from each item of the scale. There is no reverse-coded item on the scale. The lowest score that can be obtained from the scale is 21, and the highest score is 105. A high score indicates a positive attitude towards marriage, and a low score indicates a negative attitude towards marriage. As a result of the exploratory factor analysis performed to determine the scale's construct validity, it

was seen that the scale had a single factor structure that explained 36.77% of the total variance. In the analysis regarding the reliability of the scale, the Cronbach Alpha value as an internal consistency indicator was found to be .90, and the Spearman-Brown Split Test value was found to be .88 [46].

**Ambivalent Sexism Inventory (ASI).**   The scale was developed by Glick and Fiske and adapted into Turkish by Sakallı-Uğurlu [44,47]. The 22-item scale consists of two sub-dimensions: hostile and protective sexism. Both sub-dimensions consist of 11 items each. It is a 6-point Likert type (1 = strongly disagree, 6 = completely agree) measurement tool. There is no item in the scale that requires reverse coding. While the factor loads of the items in the hostile sexism factor vary between .57 and .77, the factor loads of the items in the sub-factors of protective sexism vary between .57 and .81. The Cronbach Alpha coefficient was found to be .85 for ambivalent sexism, .78 for protective sexism, and .87 for hostile sexism. In addition, the test-retest reliability coefficient of the scale was reported as .86.

### Data analysis

Data analysis was performed using SPSS for Windows 20.0 software. In the study, the relations between the variables were examined with the Pearson correlation coefficient. When the sexism variable was controlled, it was tried to determine whether sexual abuse affected the attitude towards marriage with covariance analysis.

## Results

In the study, the relationships between the variables were examined by correlation analysis, and the results are presented in Table 1.

When the correlation analysis in Table 1 is examined, it is seen that there is a statistically significant positive correlation between the attitude towards marriage and protective sexism ($r$ = .436) and hostile sexism ($r$ = .267).

In the study, the effect of sexual abuse on the attitude towards marriage was tried to be examined with covariance analysis, which allows the sexism variable to be controlled and examined. Therefore, first of all, the data were analyzed in terms of covariance analysis assumptions. In this framework, the data were analyzed in terms of normal distribution, the correlation coefficient between variables, homogeneity of variances, and characteristics of dependent and independent variables. As a result of the examinations, five extreme data that damaged the normal distribution were deleted. After this sorting process, it was concluded that the skewness and kurtosis values of the calculated data to determine the normal distribution were between -1 and +1, the variances were homogeneous ($F_{(1,716)}$ = 3.506, $p$ = .062), and the variables were suitable for covariance analysis. However, when the relations between the variables in Table 1 were examined, the hostile sexism variable was not included in the model

**Table 1. Descriptive statistics and correlation analysis results.**

|  | Attitude Towards Marriage | Protective Sexism | Hostile Sexism |
|---|---|---|---|
| **Attitude Towards Marriage** | 1 |  |  |
| **Protective Sexism** | .436** | 1 |  |
| **Hostile Sexism** | .267** | .576** | 1 |
| $\bar{x}$ | 77.35 | 39.15 | 36.27 |
| **SD** | 15.04 | 11.76 | 12.16 |
| **Skewness** | -,361 | -,137 | -,066 |
| **Kurtosis** | -,175 | -,565 | -,690 |

**Table 2. Covariance analysis results on controlling the protective sexism variable.**

| Source | Type III Sum of Squares | df | Mean Square | F | p | Partial Eta Squared |
|---|---|---|---|---|---|---|
| Corrected Model | 34640,011[a] | 2 | 17320,005 | 97,097 | ,000 | ,214 |
| Intercept | 158800,377 | 1 | 158800,377 | 890,242 | ,000 | ,555 |
| Protective Sexism | 33771,876 | 1 | 33771,876 | 189,327 | ,000 | ,209 |
| Sexual Abuse | 3825,029 | 1 | 3825,029 | 21,443 | ,000 | ,029 |
| Error | 127540,942 | 715 | 178,379 | | | |
| Total | 4457790,000 | 718 | | | | |
| Corrected total | 162180,953 | 717 | | | | |

Dependent Variable: Attitude Towards Marriage.

a. $R^2$ =, 214 (Adjusted $R^2$ =, 211).

as a control variable since the relationship between hostile sexism and attitudes towards marriage was low. For this reason, the effect of sexual abuse on the attitude towards marriage could be examined by controlling the protective gender. The result of the covariance analysis made as a result of these evaluations is presented in Table 2.

When Table 2 is examined, it is seen that protective sexism ($F$ = 189.327, $\eta^2$ = .209, $p$ = .000) and sexual abuse ($F$ = 21.443, $\eta^2$ = .029, $p$ = .000) predict attitudes towards marriage at a statistically significant level. In addition, as seen in Table 2, this finding of the study shows that the effect of sexual abuse on attitudes towards marriage is statistically significant, even without the effect of sexism (Adjusted $R^2$ = 21%). In addition, when the descriptive statistical results are examined, it is seen that the level of "attitudes towards marriage" of individuals who are not victims of sexual abuse is higher than those who are victims (Victim of Sexual Abuse = 75.87, Not Victim of Sexual Abuse = 78.17).

## Discussion

In the study, a positive and significant relationship was found between the level of marriage attitude and hostile sexism and protective sexism. When the literature is examined, no research has been found that examines the relationship between marriage attitude and ambivalent sexism. However, there are studies in the literature examining the relationships between marriage-related variables, relationship problems, relationship adjustment, and sexism [48–52]. For example, these studies showed that hostility and benevolence shape close-relationship ideals [50], benevolent sexism was negatively associated with marital quality [52], and ambivalent sexism plays a role the quality of romantic relationship [49]. The present research findings support the results of these studies.

The positive attitudes of individuals who have hostile sexism and protective sexism towards marriage may lead to the development of a perception that women should take responsibility first in matters such as ensuring a regular life in marriage, carrying out household chores, and childcare. Considering that attitudes towards marriage are shaped by society's perception of gender, gender roles imposed on individuals, and cultural values of individuals, the positive effects of hostile and protective sexist attitudes on marriage attitudes may indicate that the participants have a more traditional perspective. Glick and Fiske stated that societies with higher levels of hostile and protective sexism have lower levels of gender equality [53]. The findings also support this situation. In this context, it can be said that as a result of growing up in a culture where there is gender inequality for the individuals participating in the study, their perceptions of marriage consist of roles and expectations that are far from gender equality.

The relationship between attitude towards marriage and hostile sexism was found to be lower than that of protective sexism. Therefore, the variable of hostile sexism was not considered as a control variable. The low correlation between hostile sexism and marital attitude may be related to the fact that protective sexism is seen as more acceptable than hostile sexism and is not perceived as discrimination. In other words, since hostile sexism includes situations in which negative attitudes towards women are manifested [44], protective sexism may be more related to marriage attitude than hostile sexism, since it includes situations that express the protection and care of women. At the same time, the fact that protective sexism is associated with marriage attitude at a higher level may indicate that marriage is considered an organization where women are safe, protected, cared for, and removed from dangerous and risky situations.

Even without the effect of sexism, the effect of sexual abuse on attitude towards marriage was found to be statistically significant. According to the results, it was observed that individuals with sexual abuse experience had more negative attitudes towards marriage than individuals who did not. In the literature, when the studies on the relationship between sexual abuse experience and marriage attitude are examined, it is seen that sexual abuse experiences have a direct effect on marriage attitude [45,54]. While sexually abused individuals experience destructive emotions such as fear, anger, guilt, and hostility [19]. They also experience difficulties in many issues such as independence, closeness with others, self-care, memory, identity, and maintaining stable relationships [55]. In this respect, considering that individuals with a history of sexual abuse have problems in terms of their identity search, attachment styles, and relational expectations, it may be possible that their perceptions of marriage may also be negatively affected in this sense. On the other hand, feelings of anger, guilt, disappointment, and distrust in the individual may cause the person to avoid romantic relationships and develop a negative attitude towards marriage.

According to the results of the research, it was found that protective sexism and sexual abuse significantly predicted marital attitudes. No such study was found in the literature review. It can be thought that the effect of protective sexism and sexual abuse on the marriage attitude together is related to the need for care and protection of the sexually abused individuals more. It is stated that after the abuse act, the victim needs support from her partner, family, and environment, and the social support he/she receives accelerates the recovery process [56,57]. As support this situation, according to Hirai positive social support prevents the victim from forming negative schemas, that the self-concept is less damaged, and maladaptive coping decreases [58]. According to a study on protectionist sexism, it was found that protective sexism affects even sexual harassment as a normal flirtation [59]. Accordingly, the need for closeness and trust of individuals who have experienced sexual abuse may drag the person into relationships in which they will again enter the role of victim. The individual may not consider the protective statements presented to him/her as inequality and sexism and see his/her partner as a safe channel.

## Conclusions

The research has a certain sample limit due to cost and time constraints. At the same time, the fact that individuals who have experienced sexual abuse do not want to share it may be among the limitations. In order to ensure the generalizability of the findings of the study, it would be beneficial to conduct new studies with larger study groups and sample groups of participants from different cultural backgrounds, as well as to include other demographic characteristics in the study. However, it is thought that working with different research designs such as qualitative research will contribute to the multidimensional evaluation of the concepts of sexual

abuse, marriage attitude, and ambivalent sexism. Since childhood traumas such as sexual abuse can affect both marital satisfaction and quality of life and cause various problems in the future, it is thought that it may be beneficial to carry out studies to prevent sexual abuse and to raise awareness of parents and society. In addition, group guidance activities should be carried out to enable individuals to see the positive and negative sides of their culture and to realize their own gender stereotypes.

## Supporting information

**S1 Dataset.**
(SAV)

## Author Contributions

**Conceptualization:** Eyüp Çelik, Kübra Dombak.

**Data curation:** Lokman Koçak, Mithat Takunyacı.

**Formal analysis:** Eyüp Çelik, Mithat Takunyacı.

**Investigation:** Samet Makas, Seyhan Bekir.

**Methodology:** Mehmet Kaya, Ümit Sahranç.

**Writing – original draft:** Kübra Dombak, Mehmet Kaya.

**Writing – review & editing:** Eyüp Çelik.

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
