## [Decision Letter · Decision Letter 0]

22 May 2023

PONE-D-23-05383Examining the Attitudes of Sexually Abused and Non-Abused Individuals towards Marriage in Terms of Ambivalent SexismPLOS ONE

Dear Dr. Çelik,

Thank you for submitting your manuscript to PLOS ONE. After careful consideration, we feel that it has merit but does not fully meet PLOS ONE’s publication criteria as it currently stands. Therefore, we invite you to submit a revised version of the manuscript that addresses the points raised during the review process.

We look forward to receiving your revised manuscript.

Kind regards,

Miguel Ángel Gallardo Vigil, Ph.D

Academic Editor

PLOS ONE

Journal Requirements:

4. Please ensure that you include a title page within your main document. You should list all authors and all affiliations as per our author instructions and clearly indicate the corresponding author.

Reviewers' comments:

Reviewer's Responses to Questions

**Comments to the Author**

1. Is the manuscript technically sound, and do the data support the conclusions?

Reviewer #1: Yes

Reviewer #2: Yes

2. Has the statistical analysis been performed appropriately and rigorously? 

Reviewer #1: I Don't Know

Reviewer #2: I Don't Know

3. Have the authors made all data underlying the findings in their manuscript fully available?

Reviewer #1: Yes

Reviewer #2: Yes

4. Is the manuscript presented in an intelligible fashion and written in standard English?

Reviewer #1: Yes

Reviewer #2: Yes

5. Review Comments to the Author

Reviewer #1: The researchers have done a much needed piece of work on a sensitive subject , yet not much previously explored one.

The authors might benefit to do further studies with wider comparison in differebt regions, covering different cultural aspects.

Thank you, Surely this work will help many professionals working with the victims of secual abuse.

Reviewer #2: Greetings and Regards

Thank you for giving me the opportunity to read and peer review this manuscript. It is a valuable article and it covers an important and interesting topic.

In order to improve the quality of the manuscript, there are several things that are mentioned below.

1. The introduction section is written in great detail and it is suggested to write this section in a more concise manner and with more recent sources.

2. In Table one and the first line, numbers 1, 2, and 3 represent which variables?

3. In the conclusion section, are you sure about this statement? There are studies in this field that are below:

Ambivalent sexism in close relationships:(Hostile) power and (benevolent) romance shape relationship ideals

Ambivalent sexism and perceptions of men and women who violate gendered family roles

6. PLOS authors have the option to publish the peer review history of their article (what does this mean?). If published, this will include your full peer review and any attached files.

Reviewer #1: **Yes: **Dr Lily Abedipour MD

Reviewer #2: No

---

## [Author Response · Author response to Decision Letter 0]

13 Jun 2023

RESPONSE TO REVİEWER(S)

Manuscript: Examining the Attitudes of Sexually Abused and Non-Abused Individuals towards Marriage in Terms of Ambivalent Sexism, PONE-D-23-05383

Dear Editor/Reviewer(s),

Thank you for the valuable comments and suggestions. Your comment/suggestion is a valuable opportunity to further improve the quality of our work in terms of the technical content, novelty and quality of literary presentation.

Journal Requirements:

Comments:

Response: Article revised according to PLOS ONE's style requirements

2. Thank you for stating the following financial disclosure

Response: This statement was added in Cover Letter: 

“The funders had no role in study design, data collection and analysis, decision to publish, or preparation of the manuscript. The authors received no specific funding for this work.”

3. We note that you have indicated that data from this study are available upon request. PLOS only allows data to be available upon request if there are legal or ethical restrictions on sharing data publicly.

Response: Data Availability statement was updated in cover letter: “The data that support the findings of this study are not publicly available due to their containing information that could compromise the privacy of research participants.”

4. Please ensure that you include a title page within your main document. You should list all authors and all affiliations as per our author instructions and clearly indicate the corresponding author.

Response: A title page including list all authors and all affiliations and other informations was added.

Response: References in the text and in the reference list were reviewed and arranged according to the style of the journal.

Review Comments to the Author

Comments:

1. The introduction section is written in great detail and it is suggested to write this section in a more concise manner and with more recent sources.

Response: The introduction section is written in a more concise manner and with more recent sources. İn this context, non-essential sentences were deleted and updated references were added to the relevant sentences.

2. In Table one and the first line, numbers 1, 2, and 3 represent which variables?

Response: Variables represented by numbers (1,2,3) in Table 1 are written.

3. In the conclusion section, are you sure about this statement? There are studies in this field that are below:

Ambivalent sexism in close relationships:(Hostile) power and (benevolent) romance shape relationship ideals

Ambivalent sexism and perceptions of men and women who violate gendered family roles

Response: In the discussion section, the sentences stated by the referee were revised and current relevant research findings were added.

---

## [Editor Report · Decision Letter 1]

21 Jun 2023

Examining the Attitudes of Sexually Abused and Non-Abused Individuals towards Marriage in Terms of Ambivalent Sexism

PONE-D-23-05383R1

Dear Dr. Çelik,

We’re pleased to inform you that your manuscript has been judged scientifically suitable for publication and will be formally accepted for publication once it meets all outstanding technical requirements.

Kind regards,

Miguel Ángel Gallardo Vigil, Ph.D

Academic Editor

PLOS ONE

---

## [Editor Report · Acceptance letter]

28 Jun 2023

PONE-D-23-05383R1 

Examining the attitudes of sexually abused and non-abused individuals towards marriage in terms of ambivalent sexism 

Dear Dr. Çelik:

I'm pleased to inform you that your manuscript has been deemed suitable for publication in PLOS ONE. Congratulations! Your manuscript is now with our production department. 

Kind regards, 

on behalf of

Dr. Miguel Ángel Gallardo Vigil 

Academic Editor

PLOS ONE